# Trends and Applications of Surface and Bulk Acoustic Wave Devices: A Review

**DOI:** 10.3390/mi14010043

**Published:** 2022-12-24

**Authors:** Yang Yang, Corinne Dejous, Hamida Hallil

**Affiliations:** University of Bordeaux, CNRS, Bordeaux INP, IMS UMR 5218, F33400 Talence, France

**Keywords:** RF acoustic devices, SAW, BAW, sensor application, RF filter, physical transducers, biochemical transducers

## Abstract

The past few decades have witnessed the ultra-fast development of wireless telecommunication systems, such as mobile communication, global positioning, and data transmission systems. In these applications, radio frequency (RF) acoustic devices, such as bulk acoustic waves (BAW) and surface acoustic waves (SAW) devices, play an important role. As the integration technology of BAW and SAW devices is becoming more mature day by day, their application in the physical and biochemical sensing and actuating fields has also gradually expanded. This has led to a profusion of associated literature, and this article particularly aims to help young professionals and students obtain a comprehensive overview of such acoustic technologies. In this perspective, we report and discuss the key basic principles of SAW and BAW devices and their typical geometries and electrical characterization methodology. Regarding BAW devices, we give particular attention to film bulk acoustic resonators (FBARs), due to their advantages in terms of high frequency operation and integrability. Examples illustrating their application as RF filters, physical sensors and actuators, and biochemical sensors are presented. We then discuss recent promising studies that pave the way for the exploitation of these elastic wave devices for new applications that fit into current challenges, especially in quantum acoustics (single-electron probe/control and coherent coupling between magnons and phonons) or in other fields.

## 1. Introduction

In recent decades, the rapid development of microwave wireless communication technology has led to very rapid developments in other fields, such as mobile communication systems (CDMA, UMTS, GSM, etc.), global positioning systems (GPS, Galileo, etc.), data transmission systems (such as WLAN, Bluetooth, etc.), satellite communication, and other military communication systems [1]. The most important parts of the components that make up these systems, such as filters, duplexers, voltage-controlled oscillators, frequency meters, and tunable amplifiers, are microwave resonators [2]. With the remarkable advances in microelectronics and microfabrication technologies, researchers have paid more attention to the miniaturization and integration of devices while improving their performance. They have succeeded in integrating multiple devices on a single chip and in accommodating active and passive devices or MEMS devices in a single package [3].

Conventional microwave resonators use electromagnetic waves as energy carriers. In the RF/microwave frequency range, the required resonators have dimensions that can hardly meet the integration requirements for applications such as the current fifth-generation (5G) telecommunication standard. With wave velocities four to five orders of magnitude slower than that of electromagnetic waves, acoustic waves allow a reduction of device dimensions in the same proportions [4]. As a consequence, acoustic wave devices can meet the current miniaturization and integration requirements [5].

This paper aims to present an overview of surface and bulk acoustic waves technologies and of major application fields in a comprehensive manner, especially for the benefit of novice readers, because a profusion of literature is associated with such quickly expanding technologies. We first present the basic principles of SAW and BAW devices, typical structures, main characterizations, potential limitations, and possible optimization methods. Regarding BAW devices, we give particular attention to the film bulk acoustic resonator (FBAR), which holds a higher operation frequency and a better integration capacity compared with SAW devices. We then propose an overview of major current applications, which is divided into three parts: RF signal filters, physical sensors and actuators, and chemical and biochemical sensors. Finally, we conclude this review with a brief introduction to current trends associated with quantum acoustics and some applications.

## 2. Basic Principles of SAW and BAW Devices

Acoustic components are based on acoustic waves generated from the piezoelectric effect, one of the most exciting material properties. The term derives from the Greek word *piezein* (to press), and the effect is based on an interaction between an electrical charge and mechanical stress [6]. In 1880, the Curie brothers discovered that applying pressure or tension in a certain direction to a quartz crystal results in the formation of electrical charges on its surface, and the density of the electrical charge is proportional to the magnitude of the applied external force [7]. This is the positive piezoelectric effect of piezoelectric materials. One year later, the Curie brothers proved the inverse effect through experiments and determined the direct and inverse piezoelectric coefficients of quartz crystal [8].

Acoustic waves generated from the piezoelectric effect can be broadly classified into two types: BAW [9] and SAW [10]. Essentially, BAW are generated by an alternating (AC) electrical signal applied between the two sides of the piezoelectric material, and the acoustic wave propagates through the entire thickness, giving rise to a stationary wave at specific frequencies. Depending on the crystalline orientation, the piezoelectric material can expand and contract perpendicularly to the surface as in Figure 1a; the wave polarization is longitudinal, as the mechanical displacement is parallel to the propagation direction; such a mode is called thickness-extension mode. When the piezoelectric material deforms in a direction parallel to the surface, the mode is called thickness-shear mode, as in Figure 1b [11]. In the case of SAW, the vibrations occur only at the surface of the material. By placing interdigitated transducers (IDTs) on the surface of the piezoelectric material, as in Figure 1c, the propagating SAW will be generated when an AC signal is applied; the resonance frequency, amplitude, and propagation characteristics are determined by the design of the IDTs, the dimensions of the electrodes, the material properties, and the applied electrical signal. As for the BAW, several types of polarization and modes exist; some of the most significant ones are described in the following part.

### 2.1. Basic Structures

In 1885, Lord Rayleigh discovered the “Waves Propagated along the Plane Surface of an Elastic Solid” and defined the first discovered SAW, which were named Rayleigh waves [12]. Rayleigh waves cause surface particles to move elliptically in planes perpendicular to the surface and parallel to the propagation direction. They are typically involved in seismic movements. In 1911, Augustus Edward Hough Love mathematically predicted the existence of so-called Love waves, which can propagate when the surface of a semi-infinitely thick elastic body is covered with a layer of a medium with lower sound velocity. The Love wave is a guided shear horizontal surface wave (SH-SAW) with a displacement of the particles parallel to the surface [13]. In 1965, Richard Manning White et al. proposed to directly deposit interdigital transducers (IDTs) onto the surface of piezoelectric materials in order to generate, transmit, and receive SAW efficiently [14]. As represented in Figure 2a, in a standard SAW resonator with a delay line, when the electrical signal arrives at the input IDT (left side) through a feedline, here with matching dipole, acoustic waves are generated by the inverse piezoelectric effect and acoustic resonance occurs, at specific frequencies of constructive waves, for which acoustic waves travel across the propagation path. Arriving at the output IDT (right side), the acoustic signal is therefore converted back into an electrical signal through the output feedline by the piezoelectric effect.

Indeed, the IDT is a typical electroacoustic conversion structure in SAW filters and other SAW devices. Its basic design is shown in Figure 2b, with metal strips on the piezoelectric substrate intertwined and connected to the signal and the ground, alternately. These interdigitated electrodes are structurally characterized by the finger width *a*, interfinger distance *b*, acoustic aperture *W*, spatial periodicity *p*, and number of finger pairs *N*. When an IDT is connected to an alternating signal source, the material is deformed locally due to the converse piezoelectric effect. At the resonance frequency, the waves emitted by each pair of electrodes are constructive and the resulting SAW propagate in the two opposite directions perpendicular to the fingers. The acoustic signals reaching the second IDT are converted into electrical signals due to the piezoelectric effect. Although the two transducers are reciprocal, for simplicity, the transducer connected to the alternating signal source will be referred to as the exciting or input IDT and the transducer connected to the load will be referred to as the receiving or output IDT. The acoustic resonance frequency is expressed as
(1)$$f_{0}=\frac{v_{s}}{p}$$
and the angular frequency
(2)$$\omega_{0}=2\pi \frac{v_{s}}{p}$$
where *v_s_* is the velocity of the SAW, which depends on the piezoelectric substrate properties in the considered orientation, and the spatial periodicity *p* of the IDTs is equal to the wavelength *λ* of the propagating SAW.

Unlike SAW resonators, BAW resonators use acoustic waves which propagate in the direction of the thickness of the piezoelectric material. The thickness of the piezoelectric plate typically corresponds to half a wavelength (λ/2) of the fundamental resonance frequency of the thickness-extensional mode (Figure 3), which can be expressed as follows, assuming infinitely thin electrodes:(3)$$f_{r}=\frac{v_{L}}{2h}$$

Here *h* is the thickness of the piezoelectric layer, and *v_L_* is the velocity of longitudinal waves in the piezoelectric medium in the plate thickness direction.

Though such devices are widely used based on bulk piezoelectric materials, like clock sources and quartz crystal microbalances (QCMs), Equation (3) highlights the need for very thin plates for high frequency operation, not compatible with bulk materials. As a consequence, due to the limitations of the fabrication process, the performance of conventional BAW resonators remained lower compared with that of the SAW resonators. In parallel, piezoelectric layer-based devices have been investigated. In 1965, Newell first proposed a piezoelectric sandwich resonator structure using an acoustic Bragg reflector with thickness layers of λ/4 and indicated that it was likely to be used in the high-frequency range [15]. In 1967, Sliker and Roberts proposed a CdS-based acoustic resonator on a quartz wafer [16], and the theoretical model of the device was basically mature at this time. In 1980, Grudkowski et al. proposed the concept of a BAW resonator filter and fabricated a ZnO-based BAW filter on a Si substrate with an operating frequency of 200~500 MHz [17]. In 1981, Lakin et al. clarified for the first time the application perspectives of thin-film bulk acoustic resonators (TFBARs) [18].

With the development of microfabrication processes, the advantages of BAW resonators slowly began to appear by the end of the 20th century. In 1996, Ruby prepared a BAW resonator based on a piezoelectric AlN film with a quality factor (Q value) of 2300 and an electromechanical coupling coefficient $k_{t}^{2}$ of 6% [19]. Q value is a dimensionless ratio of the stored energy to the energy loss within a resonant element. He then investigated and prepared a 1900 MHz duplexer based on a film bulk acoustic waves resonator (FBAR) [20]. At this time, FBARs gradually began to be commercialized, which prompted more companies to conduct research on FBARs. In 2001, Agilent (i.e., Avago Avago) first introduced the PCS (personal communications systems) duplexer with an operation frequency of 1900 MHz for the mobile phone market [20], which was already in mass production at that time, officially initiating the commercial move of FBARs. The German company Infineon [21] then also launched its own BAW devices. Later, Intel [22], TriQuint [23] in the United States, Philips [24] in the Netherlands, and Samsung [25] in South Korea joined the development of BAW resonators.

Conventional QCM type BAW devices have been well-investigated, with a variety of fundamental and harmonic modes. We focus on the emerging thin-film-type FBAR, with some outstanding features compared with the classical BAW and even SAW. The current FBAR devices can be divided into two main types by their structures: the front-side etch or airbag type and the solidly mounted type (solidly mounted resonators; SMRs). Both are based on the same principles; the main difference is the method of energy confinement. In the FBAR, an air cavity with a length and width of 100–300 μm is etched under the bottom electrode in order to obtain a suspended piezoelectric membrane confining the acoustic energy, as illustrated in Figure 4a [26]. In the SMR structure, a “mirror” under the electrode reflects acoustic waves. As shown in Figure 4b, these acoustic Bragg reflectors alternate layers of different acoustic impedances, such as W and SiO_2_ (impedance ratio of about 4:1), AlN and SiO_2_ (impedance ratio of about 3:1). By reflecting acoustic waves back to the piezoelectric film, they play a role in limiting energy dissipation [27]. To design the acoustic Bragg reflectors, the acoustic impedance of each material layer can be calculated as follows:(4)$$Z_{a}=\rho v_{L}$$
where *ρ* is the material density and *v_L_* is the velocity of the longitudinal wave in the film thickness direction if considering the SMR thickness-extensional mode.

The differences in structure and acoustic reflection of the two resonators determine the differences in their fabrication, performance, and applications. In terms of design and fabrication, the longitudinal acoustic waves are confined in the FBAR membrane, the design is flexible, and the processing is quite simple due to the smaller number of layers [28]. In the design of SMR, not only does the lateral acoustic energy dissipation need to be considered (the impact brought by the lateral acoustic modes is presented below), but the design of the acoustic Bragg reflector is also critical as it directly impacts the energy dissipation and the performance of the resonator. To ensure the Q value, a fine control of the thickness during processing is also required [26].

As for performance, FBAR has a higher effective electromechanical coupling coefficient ($k_{t}^{2}{}_{eff}$) [28], with a larger difference in acoustic impedance at the electrode–air interface than at the junction with the Bragg reflector of the SMR structure. Furthermore, the Bragg reflector adds some new loss mechanisms that reduce its Q value. However, the SMR structure also offers some interests. Indeed, the multiple SiO_2_ layers of the Bragg reflector have a negative temperature coefficient of frequency (TCF), which provides a matching effect on the TCF of the whole SMR [26]. Second, the FBAR membrane is mechanically supported at the edge of the cavity structure, which is risky during the microfabrication process, and the stress of the film must be carefully controlled to avoid mechanical cracking. On the contrary, the SMR structure does not suffer from such drawbacks, provided there are stable layer interfaces [9].

### 2.2. Typical Characterizations

SAW devices are usually electrically characterized by their scattering parameters (S-parameters), *S_ij_* corresponding to the output power at port *i* divided by the input power at port *j*, or transmission gain from port *j* to port *i*.

Figure 5 gives an example of the characterization results of a SAW delay line (two-port device) such as those used by Rubé, M et al. [29], measured with a vector network analyzer (VNA) in the frequency domain. It exhibits a resonance at about 118 MHz, with a sharp dip in the reflection parameter *S*_11_, which corresponds to a maximum of *S*_21_, the transmission being supported by an acoustic wave between ports.

Among other applications, signal filtering is one of the most typical applications for SAW devices, for which they are designed to meet frame specifications, in terms of passband and stopband, as well as maximum and minimum losses inside, respectively.

In the above, we have presented the classic design of SAW devices. However, reflections on metal electrodes or other interfaces, material losses, and other spurious modes can highly influence the performances. Such effects can be reduced by an appropriate design, as described here.

The first important effect is the internal reflection at the metal strips of an IDT. This is illustrated in Figure 6a, with an electrode pitch equal to half the center frequency wavelength (*λ*_0_/2) resulting in an in-phase addition of the unwanted reflections at this frequency, which causes a significant impact on performances by causing ripples in the amplitude and phase. With a split-finger IDT, as shown in Figure 6b, the electrode pitch becomes *λ*_0_/4, so that the reflections between two adjacent electrodes cancel each other, being 180° out of phase, and the overall reflected wave is cancelled.

In some designs, gratings on the surface are used to improve the confinement of acoustic energy in the transducer and consequently the Q value of the SAW device. In Figure 7, a one-port resonator has two such metallic side gratings, which act as two reflectors for the acoustic energy generated by the central IDT, therefore improving the Q value. Metallic gratings can not only provide mechanical but also electrical reflection, since an electrical field can also be formed by reflected acoustic waves due to the piezoelectric effect. The electrical characteristics of these reflections depend on the metal strips’ electrical connections, either short-circuited as in Figure 7, or open-circuited, or a combination of them which is named as positive and negative reflection (PNR) grating, with enhanced reflection properties. Gratings are also widely used in two-port SAW delay-line structures [30]. Another way of limiting bidirectional losses lies on specific designs enabling control of internal reflections within the generating IDT itself, such as in a single-phase unidirectional transducer (SPUDT) [4].

Spurious BAW modes are also a possible source of reduced performance. Among them, so-called deep bulk acoustic waves (DBAW) can be generated by the input IDT, then propagate in the volume of the piezoelectric layer, get reflected at the bottom face, and propagate back to the receiving IDT. Surface skimming bulk waves (SSBW) and other leaky waves are also examples of possibly interfering (or sometimes alternating) waves [31].

Waves undergoing reflections on the device sides or bottom can be limited by sand-blasting the bottom surface or biasing the sides in order to avoid constructive reflections travelling back to the IDTs. Other interfering waves such as SSBW need, for intrinsic good design, to take into account both longitudinal and shear acoustic waves.

Other particular IDT structures can also be implemented during the design process in order to achieve a specific response pattern, such as apodization [32], weighted IDT transducers [33], multistrip couplers (MSCs) [34], etc.

All these optimizations aim to reduce the impacts of spurious modes or secondary effects, to better confine the acoustic energy and improve the Q value, and therefore to improve the performances of SAW devices.

The BAW devices have a single port; they are usually characterized by measuring their reflection coefficient *S*_11_. As shown in Figure 8a, a minimum return loss is obtained near the resonance loss with a high Q value. For a BAW device with good performance, the Q value can reach several thousand [19,35,36].

In addition to *S*_11_, the electrical impedance resulting from the theory of transmission lines [37] is an important characterization property for BAW devices, which can be expressed as
(5)$$Z=\frac{1+S_{11}}{1-S_{11}}Z_{0}$$
where *Z* is the electrical impedance of the device and *Z*_0_ is transmission line characteristic impedance, typically 50 Ω for a vector network analyzer (VNA).

As is shown in Figure 8b, the electrical impedance *Z* reaches a minimum (tends to zero) at the resonance frequency *f_r_*, which corresponds to a maximum of the mechanical deformation caused by the piezoelectric effect; at the antiresonance frequency *f_a_*, the electrical impedance *Z* reaches a maximum (tends to infinity).

However, longitudinal deformations are accompanied by transverse ones, so that some lateral acoustic modes also propagate in BAW devices [38]. These unwanted acoustic modes, or spurious modes, are visible in the electrical response of the resonator in the form of parasitic resonances, in addition to the main one. These lateral acoustic waves travel between the boundaries of the active region of the piezoelectric layer, bounce off the electrode edges, and form lateral standing waves. Since they have to share the total energy, they are responsible for the degradation of the effective electromechanical coupling coefficient and of the Q value [26].

There are mainly two kinds of methods to improve this degradation caused by undesirable lateral modes. One method is called apodization. By using an asymmetrically shaped top electrode [39], most of the standing lateral waves are smeared out between the electrode edges and fewer parasitic resonances are observed in the electrical response. The most commonly used shapes of BAW top electrodes are irregular squares, pentagons, and circles. Another method is to build a frame around the edge of the top electrode [40]. By carefully designing the width and thickness of the frame, the different orders of lateral modes couple together and are vanished. This is an efficient method for suppressing unwanted modes, confining energy, and obtaining a smooth electrical response and a better Q factor [40]. However, due to the complexity of this structure and the difficulties in processing, the apodization method is commercially very successful.

## 3. Current Applications

### 3.1. RF Filters

SAW devices are widely used for signal filtering in the field of telecommunication. To meet the current requirements, filters must have a large enough passband, which can be adjusted with a suitable IDT design. In this perspective, as for unidirectional IDTs, an appropriate design can generate specific filter templates [4]. Beyond that, much attention has also been paid to filter configurations that smartly combine several one-port SAW resonators, as impedance elements, with different topologies such as interdigitated interdigital transducer SAW (IIDT), double-mode SAW (DMS), or ladder-type [41]. The ladder-type filter is a very common configuration of such low-loss coupled SAW, represented in Figure 9, which consists of cascaded multiple stages, each one based on two SAW resonators connected in series and in parallel. Coupling them by matching the resonance frequency of a serial resonator with the antiresonance frequency of the parallel one results in a bandpass filter centered on this frequency. Such filters exhibit a relatively flat passband with low loss and good rejection of out-of-band noise [30]. Similarly, a design coupling identical SAW resonators in double symmetric and antisymmetric modes by inverting their electrical connections can result in a wider passband filter [30,42].

There are some important properties to characterize a SAW filter:

(1) The minimum insertion loss (*S*_21_ parameter), quantifying the power dissipation caused by the device access, which depends on the input/output impedances of the device itself and of the input and output circuits. In case of unmatching, calibration tests are used to post-process the measurement results. Acoustic propagation can also participate in losses.

(2) The center frequency, the arithmetic average of the two cut-off frequency values, −3 dB or half-power of the minimum insertion loss level.

(3) The nominal frequency range, which is the usable bandwidth over which signal transmission is observed, defined as the range between the two cut-off frequencies.

(4) The out-of-band rejection, the ratio of signals inside and outside the passband, which is defined as the drop-off value between the edge of the passband and the maximum value of the stopband.

(5) The Q value determines the maximum intrinsic bandwidth of a filter; it corresponds to the ratio of its center frequency to its 3 dB bandwidth.

With the rapid development of 5G technology, RF filters with a high frequency and Q value become increasingly in demand in the mobile phone industry as well as with new growth opportunities. This context gives RF SAW filters a significant market prospect. From the report “Surface Acoustic Wave Filter Market” by Persistence Market Research, the global SAW filter market registered a compound annual growth rate (CAGR) of 7.5% between 2015 and 2020, and was expected to reach USD 5 billion in 2021 with a CAGR of 9% by 2031. Over 50% of the market is shared by American and Japanese companies, such as Qualcomm Technologies, Qorvo, Skyworks Solutions, Microchip Technologies, and Murata Manufacturing [43,44].

Like SAW, BAW devices, currently increasingly of FBAR type, are also fundamental components for RF filters, requiring a wide bandwidth with low insertion loss and a stopband enabling the suppression of unwanted signals. Similarly, as for the SAW components, the ladder-type filters are also commonly used due to advantages such as easy design, steep filtering effect, cost, etc. Some communication systems mix ladder filters and lattice ones, in which shunt elements are diagonally crossed, to achieve a good selectivity of frequency bands with steeper filtering response [26,45]. Since the resonance frequency of SAW mainly depends on the spatial periodicity of IDTs, which is limited by lithography and patterning technology, it is quite difficult for SAW to operate above 2 GHz [46], so FBARs usually dominate the market for filters above 2.5 GHz. This is further reinforced with the 5G New Radio (NR) systems, since a main feature is the use of high-frequency millimeter wave (mmWave) and sub-6 GHz bands [47]. As a consequence, the operating frequency band will continue to expand to high frequencies, and the working bandwidth will also increase, which can be supported by the newest FBAR-based BAW technologies.

As a result, the market for filters is expected to grow explosively. Among them, the growth of BAW filters is the fastest. Indeed, the demand for connected devices, such as vehicles, is leading to the new adoption of interface standards such as Wi-Fi, and BAW filters can also be used to establish a mobile connection with a network to enable a next-generation driving experience. The applications extend not only to the consumer electronics and automotive industries, but also to aerospace, defense, environment and industry, etc. These broad industrial applications provide prospects for significant and stable growth to the BAW filter market, which was estimated at USD 4.1 billion in 2020. By 2027, the global market for BAW filters is expected to reach USD 13 billion with a CAGR of 18% [48]. In terms of regional analysis, according to Maximize Market Research, Asia-Pacific has the largest market share and will continue to hold it in the future; Asian countries such as China, India, South Korea, and Japan are also the main consumers, and China holds the most important part due to its mature semiconductor manufacturing, telecommunications, and electronics industries [49]. The top five manufacturers in the BAW filter market are Avago Technologies (USA), Qorvo (USA), TDK (Japan), Skywork Solutions (USA), and Akoustis Technologies (USA) [49].

### 3.2. Physical Sensors and Actuators

Beside the filtering application, SAW and BAW devices are also widely used as sensors and different configurations have been developed for various physical monitoring. Among them, the QCM has been used as a way of real-time monitoring of thin film deposition thickness in microelectronics such as for evaporation equipment, based on mass-effect, with a resolution in the order of ng·cm^−2^ [50]. Many other applications have been studied, such as magnetic field [51], pressure and temperature [52], acceleration [53], tire–road friction [54], and gyroscopes [55], in many sectors including automobile, consumer, etc. [56]. In addition, SAW devices are also reported as motors and actuators [57].

A SAW resonator with a delay line is reported as a highly sensitive magnetic-field sensor by several studies [58,59,60]. For example, Meyer et al. [58] reported a thin-film-based SAW magnetic-field sensor with a limit detection of $2.4\;\mathrm{nT}/\sqrt{\mathrm{Hz}}$ at 10 Hz and $72\;\mathrm{pT}/\sqrt{\mathrm{Hz}}$ at 10 kHz. This magnetic-field sensor is composed of an AlScN piezoelectric layer, AlCu IDTs, a SiO_2_ smoothing (guiding) layer, and a magnetostrictive FeCoSiB film deposited on the delay-layer area. In the study of Schmalz et al. [60], a multimode Love-wave SAW magnetic-field sensor was designed and the first- and second-order Love-wave modes showed sufficient sensitivity. This sensor is composed of an ST-cut quartz as the piezoelectric layer, a SiO_2_ layer, and a magnetostrictive (Fe_90_Co_10_)_78_Si_12_B_10_ film on the delay-line area. The BAW-based magnetic-field sensor is less used, but some designs and simulations of the BAW-based magnetic-field sensor are reported [61,62].

SAW devices are also good candidates for pressure and temperature sensing [63,64,65,66]. In the Rodríguez-Madrid et al. [66] study, a SAW-based pressure sensor was reported with a sensitivity of 0.33 MHz/bar, a working frequency between 10 to 14 GHz with high-order harmonic acoustic modes. It is a one-port resonator with AlN as the piezoelectric layer deposited on a free-standing nanocrystalline diamond (NCD) layer. Müller et al. [65] reported a SAW-based temperature sensor with a sensitivity higher than 300 kHz/°C with an operating frequency around 5.4 GHz. It is a GaN-based SAW sensor; the detection of temperature is realized by tracking resonance frequency changes as a function of temperature. Many other studies have also reported SAW- and BAW-based sensors for high temperature detection, suitable for operation in harsh environments [67,68,69].

At the same time, SAW-based mechanical sensors are widely used in the automotive industry, with applications such as acceleration, tire–road friction sensors, etc. In this case, a wireless readout is often used as very useful facility [52,56]. For example, Wen et al. [53] reported a SAW-based acceleration sensor with a sensitivity of 29.7 kHz/g. It is a two-port SAW device with an operating frequency of 300 MHz and a very good temperature compensation system achieved by using a metal package base. SAW sensors are also used for tire pressure and tire–road friction in car and truck tires as illustrated in this study [54]. A continuous monitoring of tire pressure, which can be reduced to 50 mBar, is achieved to estimate the riding conditions (for example with braking maneuvers, over a curbstone, etc.).

SAW devices have also been well-investigated as gyroscopes for several decades [70]. For example, a standing-wave-type SAW gyroscope was proposed by Kurosawa et al. [71]. This design was then confirmed by Varadan et al. [72] with experimental results. A two-delay-line SAW micro rate gyroscope was then proposed by Lee et al. [73]. Another progressive-waves-type SAW gyroscope was proposed by Oh et al. [74,75].

Besides such physical sensing applications, SAW devices can also serve as motors and actuators [57]. For example, Kurosawa et al. [76,77] proposed a Rayleigh-type SAW-based motor with an operating frequency near 10 MHz. It is a two-port SAW device with a delay line; a preloaded slider is placed on the wave propagation path and is driven by the frictional force. In addition, SAW devices are also widely used in microfluidic actuation and micro-object manipulation [78], based on travelling or standing waves between two sets of IDTs. Leaky-type travelling waves are usually used for actuation and manipulation, and Lamb-type standing waves are used for micro-actuators [79].

### 3.3. Biochemical Sensors

Apart from physical sensors, biochemical sensing has become a popular research topic, and a great deal of work has been carried out related to such applications of SAW and BAW for gas and liquid media [80]. Currently, such sensors can show a sensitivity with detection limits down to the ppb range [81], and SAW devices also receive intense attention in biosensing for their high sensitivity, high efficiency with label-free detection, real-time monitoring, and relatively low cost [82,83]. The sensitivity is a key factor characterizing the SAW sensor’s performance [52], often improved with a sensitive layer deposited on the acoustic propagation path, which is supposed to immobilize or interact specifically with target molecules. As the acoustic wave propagation is perturbed, both the resonance frequency and the minimum loss (typically *S*_21_ attenuation for a two-port SAW device such as a delay-line device) are modified due to an additional mass or viscoelastic property changes at the near surface. By tracking *S*_21_ attenuation, resonance frequency shift, and phase, various vapors or liquids with different concentrations can be distinguished. Additional information on the real-time behavior of the adjacent medium can even be extracted from the electrical characterization out of acoustic resonance or on the reflection parameters [29].

Indeed, in 1979, Wohltjen and Dessy [84,85] first demonstrated the application and possibility of chemical/gas sensors based on SAW devices. Since then, by depositing different sensitive layers, SAW device-based gas sensors have been developed for detection of H_2_ [86], H_2_S [87,88], NO_2_ [89,90], CO_2_ [89], CH_4_ [91], SO_2_ [92], NH_3_ [93], O_3_ [94], O_2_ [95], CO [96], volatile organic compounds (VOCs) [97,98,99], explosives [100,101], etc. In 2022, Singh et al. [102] reported a SAW-based particulate matter (PM2.5) sensor which is wearable and shows a high detection sensitivity. Another important advantage of SAW device-based sensors is that they are passive components with the potential to be interrogated wirelessly. Indeed, by using antennas, acoustic waves can be excited and received by RF electromagnetic signals. This allows SAW-based gas sensors to work in high-temperature, high-pressure, and toxic environments. Wen et al. [103] reported such a wireless SAW gas sensor with Teflon AF as a sensitive layer for CO_2_ detection; they reported a sensitivity in phase shift of 1.98°/ppm. Later, Lim et al. [89] developed a remotely controlled SAW sensor for the detection of CO_2_ and NO_2_ with simultaneous temperature measurement. The sensitive layers for CO_2_ and NO_2_ are Teflon AF and indium tin oxide, and the sensitivities are 2.12°/ppm and 51.5°/ppm, respectively. Xu et al. [104] developed a wireless SAW sensor for organophosphorus compound detection; the sensor exhibits good linearity and repeatability, and a sensitivity of 20.1°/(mg/m^3^).

Apart from applications in a gaseous environment, SAW-based sensors are also widely used in the liquid phase. In this case, they mostly involve waves horizontally polarized, as vertical components suffer from fast attenuation in the liquid phase, and SAW Love-mode is actually well-investigated for its high sensitivity of detection, especially in liquid [105]. Among important applications of such biosensors is the detection of deoxyribonucleic acid (DNA). Y. Hur et al. [106] reported a 15-meroligonucleotide DNA sensor in liquid solutions with a sensitivity reaching 155 ng/mL/Hz. Kim et al. [107] reported a DNA sensor with a low detection limit of 1 ng/mL and rapid response; this sensor has the potential to be used in wireless mode. Zhang et al. [108] developed a DNA sensor with a sensitive layer of deoxy-nucleoside transferase in order to increase the phase shift, thus lowering the detection limit down to 0.8 pM. Cai et al. [109] reported a DNA sensor with a high-order harmonic acoustic mode; the sensitivity can reach 6.7 × 10^−16^ g/cm^2^ for target DNA. SAW devices are also used for protein detection. Agostini et al. [110] developed a biosensor targeting the Streptavidin protein and the detection limit is down to sub-nanomolar, 104 × 10^−12^. M. Choi et al. [111] developed a SAW sensor for cardiac troponin I; the detection limit is down to 24.3 pg/mL. Zhang et al. reported a carcinoembryonic antigen (CEA) biosensor; the sensitivity was reinforced by injecting a gold staining solution, allowing a detection limit down to 1 ng/mL. Jandas et al. [112] also reported a CEA sensor; the delay-line area was coated with gold and immobilized self-assembled monolayers (SAMs) of anti-CEA antibodies. The detection limit is down to 0.31 ng/mL. They later improved their sensor with a nanomaterial thin-film bioreceptor and the detection limit reached down to 0.084 ng/mL [113]. Apart from the detection of proteins, Brugger et al. [114] reported the use of SAW for monitoring the formation of neural networks and some investigation for real-time monitoring of living matter, such as organoids or other biomaterials, along with the addition that innovative nanodrugs for efficiency and/or toxicity evaluation can also be imagined [115]. Since the COVID-19 pandemic began, some SAW-based sensors for COVID virus detection have also been reported [116,117].

Similarly, BAW devices are also widely used as chemical and biological sensors, firstly based on the classical QCM, and taking into account surrounding physical parameters such as temperature and pressure, as previously described. Its operating frequency can reach up to tens of MHz and thickness-shear mode (TSM) is mostly studied [86]. With similar advantages to SAW devices but a lower frequency, some commercial products based on QCM allowed use for a large number of applications involving mass change measurements at the nanoscale resolution. Moreover, the addition of dissipation monitoring, known as “QCM-D”, allows an improved characterization of both mass and viscoelastic properties changes of the medium driven at the near surface [118]. With the rapid development of BAW devices, the FBAR is also becoming a popular topic due to its good sensitivity with a higher operating frequency ranging from several to tens of GHz. The detection field includes mass pressure, gas, liquid, chemical/biosensor, etc. [119] For example, Chen et al. [120] developed a ZnO-based FBAR as a gravimetric DNA biosensor with a working frequency of 1.67 GHz. As for SAW sensors, a sensitive layer is deposited onto the top electrode of the device and the absorbed target compounds interfere with the generation and propagation of the acoustic waves. Again, since the vertical deformation is strongly attenuated into an adjacent liquid medium, a thickness-shear-mode resonator (TSM) is usually preferred for biochemical sensing applications [121].

Since the FBAR is a one-port device, detection typically consists in tracking the attenuation changes as well as the frequency or phase shift of the *S*_11_ parameter at the resonance [120]. The detection is also based on mechanical effects, among them the mass loading effect. Compared to the SAW sensor, whose operating frequency is typically in the range of one hundred MHz to GHz, the operating frequency of the FBAR sensor is usually in the range of sub-GHz to about 10 GHz due to the wave confinement in a very thin layer, which provides the FBAR sensor with a high sensitivity. However, a high Q factor should be ensured to accurately detect small frequency shifts.

In the field of gas sensing, Lin et al. [122] developed an FBAR-based sensor exhibiting a high sensitivity for trinitrotoluene (TNT) and 1,3,5-trinitro-1,3,5-triazacyclohexane (RDX). The detection of hydrogen (H_2_), carbon monoxide (CO), and ethanol vapors was also reported by Benetti et al. [123], with detection limits of 2 ppm, 40 ppm, and 500 ppm, respectively. Coupling an FBAR-based sensor with a micro-preconcentrator, Yan et al. [124] showed a high sensitivity for dimethyl methyl phosphonate (DMMP), down to 2.64 ppm, with a fast response and a short recovery time. Zeng et al. [125] developed a temperature-compensated film bulk acoustic wave resonator (TC-FBAR) functionalized with a bilayer self-assembled poly (sodium 4-styrene-sulfonate)/poly (diallyl-dimethyl-ammonium chloride) to detect and identify volatile organic compounds (VOCs), with an interesting approach based on temperature modulation as a multiparameter virtual sensor array. Gao et al. [126] also proposed a solution for VOC identification, based on a dual transduction using mass and resistance variation of a conductive polymer, poly (3,4-ethylenedioxy-thiophene) and poly (styrene sulfonate) (PEDOT: PSS), deposited on the top of the device. The detection of 380 ppm of methanol was reported.

An FBAR device is also a good candidate for sensing in the liquid phase. As the aging of the population and disease concerns have been major topics for many decades, not to mention the impact brought by the novel coronavirus (COVID-19), biosensing technology in the liquid phase based on FBAR devices shows a wide perspective. The first FBAR-based biosensor was reported in 2003 [127]. Clear frequency shifts show DNA attachment and protein coupling. In 2004, Gabl et al. [128] reported a ZnO-based FBAR biosensor with a working frequency up to 2 GHz for the detection of DNA and protein [128]. In 2006, Weber et al. [129] showed experimentally that in the liquid phase, shear-mode FBARs have much better performance than longitudinal-mode FBARs due to higher quality factor (Q value) and lower noise level, as expected in the liquid phase. DNA sequences were also successfully detected by Zhang et al. [130], using a gold-top electrode FBAR and monitoring the shift of resonance frequency when DNA hybridization occurred. In 2011, Auer et al. reported DNA detection in a diluted serum (1%) with a resolution of 1 nM. Apart from the detection of DNA, FBAR devices have also shown their capacity to detect prostate-specific antigen (PSA), alpha-fetoprotein (AFP), and CEA [131]. Previously in 2011, Lin et al. [132] reported an FBAR sensor with a detection limit of PSA of 25 ng/cm^2^. Zhao et al. [133] reported a sensor of human prostate-specific antigen (hPSA) with a sensitivity of 1.5 ng/cm^2^. Chen et al. [134] reported a sensor of AFP with a detection limit of 1 ng/mL. Zhang et al. [131] developed a sensor of CEA, and the detection limit ranges from 0.2 to 1 mg/mL.

As we presented above, acoustic wave device (SAW and BAW)-based sensors have wide applications as physical and biochemical sensors; the market for these sensors also holds an important share. Based on the report “Acoustic Wave Sensor Market—Forecasts from 2021 to 2026” by Research and Markets, the global acoustic wave sensor market will grow from USD 836.17 million in 2019 to USD 2400.54 million in 2026, with a CAGR of 16.3%. The major players are Hawk Measurement Systems (Melbourne, Australia), NanoTemper technologies GmbH (München, Germany), Pro-micron GmbH (Kaufbeuren, Germany), Siemens (Munich, Germany), and Transense (Oxford, UK) [135].

## 4. New Trend: Quantum Acoustics

The discussions above proposed a review of SAW and BAW devices and key applications, up to the most recent ones in terms of major trends, namely high-frequency filters and sensors. Here, we wish to give a brief introduction about a novel stream of application: quantum acoustics. Over recent decades, the research on electrons, photons, or magnetics has led a brilliant revolution in both scientific and industrial fields, and in recent times, attentions have shifted to the phonon, a quasi-particle that describes the excitation and vibration in a periodic, elastic arrangement of atoms or molecules [136]. Since most acoustic devices employ the mechanical vibrations generated from the piezoelectric effect, acoustic devices are ideal candidates to probe and control these quantum excitations in certain conditions. This is especially true for SAW devices, since their detection and sensing applications are well developed, and some examples are reported in this part.

### 4.1. Single-Electron Probing/Controlling

With the development of semiconductor technology, current integrated circuits are composed of a huge number of transistors. In order to lower the power consumption while improving the performance, scientists have put a large amount of effort into the field of low-dimensional electronic conductors to single-electron electronics in recent decades. Recently, SAW devices have shown good potential for single-electron probing or controlling due to their high operating frequencies and high Q value at low temperatures [137,138,139]. In order to detect the coupling of single-electron and SAW, a single-electron transistor (SET) is necessary [140]. Compared to classic transistors, with advantages in terms of size, voltage, and sensitivity, the SET is the most sensitive electrometer, allowing single-electron control with working temperatures in the mK range.

Gustafsson et al. [137] presented local probing of SAW for the detection of single electrons, reaching the single-phonon level at a frequency of 932 MHz. For the sample layout, a SET was placed on the propagation path of a SAW resonator, which was polarized by the piezoelectric charge when SAW passed underneath. The measurement setup was well designed and SAW were detected by their rectifying effect on the SET’s gate modulation curve. Propagating acoustic pulses with an extremely low magnitude were detected. After calculation, for each pulse, the SAW energy passing under the SET was less than a single phonon on average, which proved the possibility of single-shot phonon detection.

Other applications in single-electron controlling are also reported. For example, Takada et al. [138] reported a SAW-driven single-electron transfer with an efficiency of 99%, which can be used to perform quantum logic operations with flying electron qubits and is a significant step to efficient quantum computers. Hsiao et al. [141] used a SAW-driven lateral *n* − *i* − *j* junction, which operates in the single-electron limit, to generate single photons, and this electron-to-photon quantum transfer marks the first step for long-distance qubit transfer.

### 4.2. Coherent Coupling between Magnons and Phonons

As we showed above, the coupling of photon–phonon is proven in certain conditions and some SAW-driven devices are developed for single-electron controlling. Therefore, magnetic and acoustic excitations, magnons and phonons, are also expected to interact with each other; in particular, magnons are shown to undergo a strong coupling with microwave photons [142,143,144]. For this reason, increasing attention is focused on coherent interactions between magnons and phonons recently [145,146,147,148].

Weiler et al. [148] demonstrated the detection of acoustic-driven ferromagnetic resonance (FMR), which shows the magnon excitation induced by SAW. Figure 10a shows the experimental setup, with a 50 nm thick polycrystalline ferromagnetic nickel film deposited on the propagating path of a SAW resonator. On Figure 10b is represented the *S*_21_ parameter without an applied external magnetic field; several resonances can be observed clearly and the inset highlights a 5 dB magnitude difference of the ninth harmonic resonance under two different magnetic fields. Figure 10c shows the SAW delay-line transmission parameter as a function of magnetic field strength; it exhibits a valley associated with Ni FMR, therefore proving the magnetoelastic coupling.

Recently, Zhao et al. [149] succeeded in visualizing acoustic FMR by micro-focused Brillouin light scattering (*μ* − *BLS*). Casals et al. [150] showed independent imaging of magnons and SAW with the synchrotron X-ray source. Besides these examples of quantum acoustics applications, the strong coupling between magnons and phonons is still under investigation [146,151,152,153,154]. Furthermore, research in other fields continues, such as coherent coupling between phonons [155,156,157,158], coupling between elastic waves and single quantum dots [159,160,161,162], etc. These advances in quantum acoustics will likely bring a revolution in current electronics science and engineering.

## 5. Conclusions

In this review, we provided a global view about current acoustic devices. We presented the piezoelectric effect, basic structures, the acoustic theory of SAW and BAW components, and gave more details for thin-film-type BAW resonators or FBARs, which hold interesting features in terms of frequency operation and integrability compared with conventional QCM devices. We also presented the possible spurious modes and some optimized designs to reduce them and therefore improve the response. Acoustic devices have been developed over the past decades; they have proven their wide applications in communication systems. Today, with the rapid development of the fifth generation (5G) and telecommunication standards, these acoustic devices, especially FBARs, represent a broad market as RF filters, compared with conventional electromagnetic devices, thanks to much slower propagation velocity allowing for shorter wavelength and, thus, easy miniaturization and integration into circuits. We then presented another important field of applications of SAW and BAW/FBARs, namely as sensors and actuators. A section was dedicated for their application as physical sensors. Examples of their use for magnetic field, pressure, and temperature monitoring and detection were illustrated. In addition, their application in other fields such as mechanical (in automotive) and orientation measurements were presented. Some examples of SAW-based motors and actuators were also introduced. We then focused on SAW/BAW-based biochemical sensors, which are receiving increasing attention in the research field. Indeed, because of their performances, among them a high sensitivity, a versatile feature that makes them easily functionalized for selectivity, and low cost, they are widely used for gas, liquid, bio-sensing, etc. The sensing applications are still under development, with a rising demand especially for biosensors, since health concerns are more than ever a major topic. As of now, SAW and FBAR devices show a very good capacity for sensing DNA, RNA, proteins, and a wide variety of other bio-compounds. With the COVID-19 pandemic, several biosensors based on SAW and FBAR devices are also reported for the detection of SARS-CoV-2 virus and application for living-matter monitoring is under development, which could be helpful for fast screening of therapeutic nanodrugs, for example. Lastly, we presented current trends related to quantum acoustics, which studies the behavior of phonons and their interactions, as opportunities for new schemes to control quantum information and explore atomic physics beyond photonic systems. SAW is the ideal candidate in this emerging field with interest in both fundamental and applied research.

## Figures and Tables

**Figure 1 micromachines-14-00043-f001:**
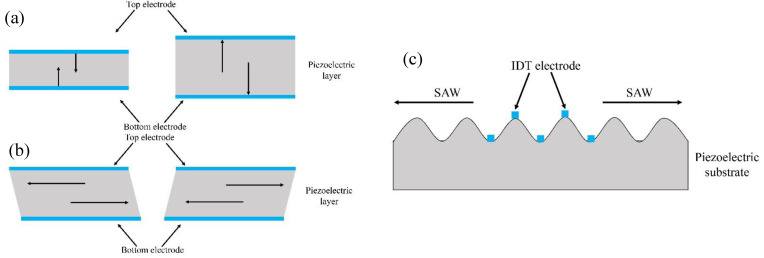
Profiles of (**a**) thickness-extensional mode and (**b**) thickness-shear mode of a BAW device; (**c**) SAW.

**Figure 2 micromachines-14-00043-f002:**
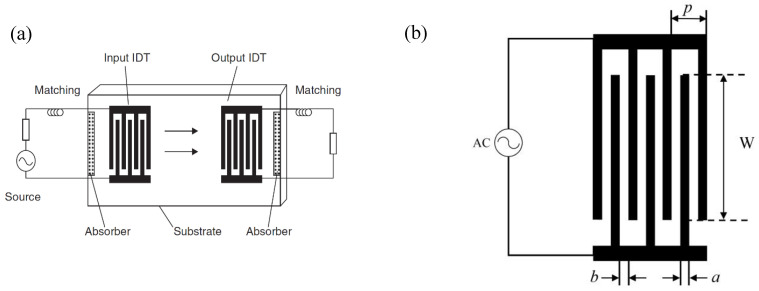
(**a**) Schematic diagram of classic SAW resonator with delay line, adapted with permission from [4]; (**b**) the classic structure of IDT.

**Figure 3 micromachines-14-00043-f003:**
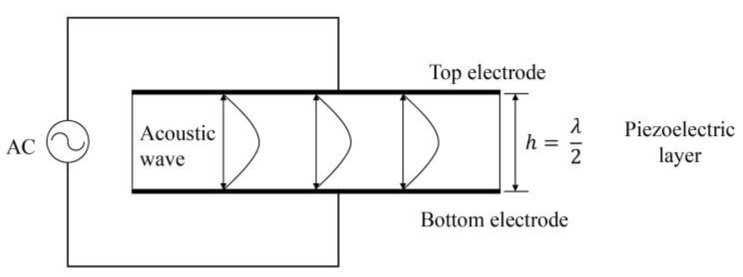
Schematic diagram of BAW resonator.

**Figure 4 micromachines-14-00043-f004:**
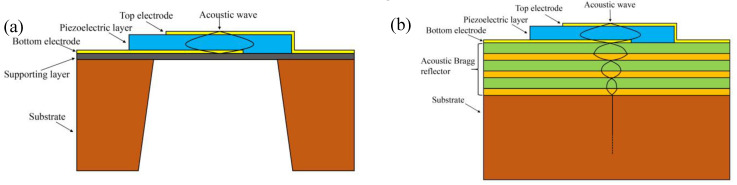
(**a**) Conventional FBAR structure; (**b**) SMR-type FBAR structure.

**Figure 5 micromachines-14-00043-f005:**
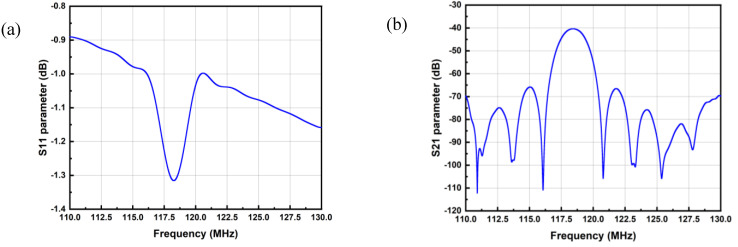
(**a**) *S*_11_ parameter; (**b**) *S*_21_ parameter of typical SAW resonator frequency response.

**Figure 6 micromachines-14-00043-f006:**
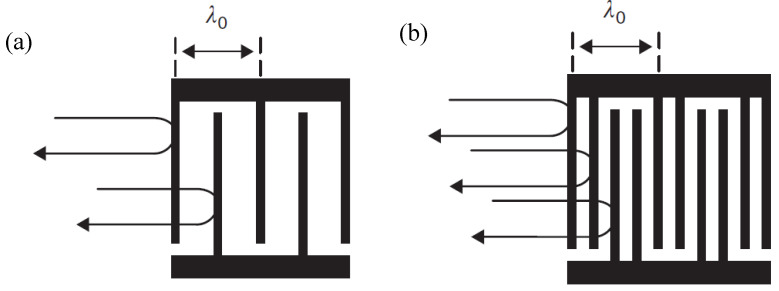
The reflection of SAW (**a**) single-electrode IDT; (**b**) split-finger electrode IDT, adapted with permission from [4].

**Figure 7 micromachines-14-00043-f007:**
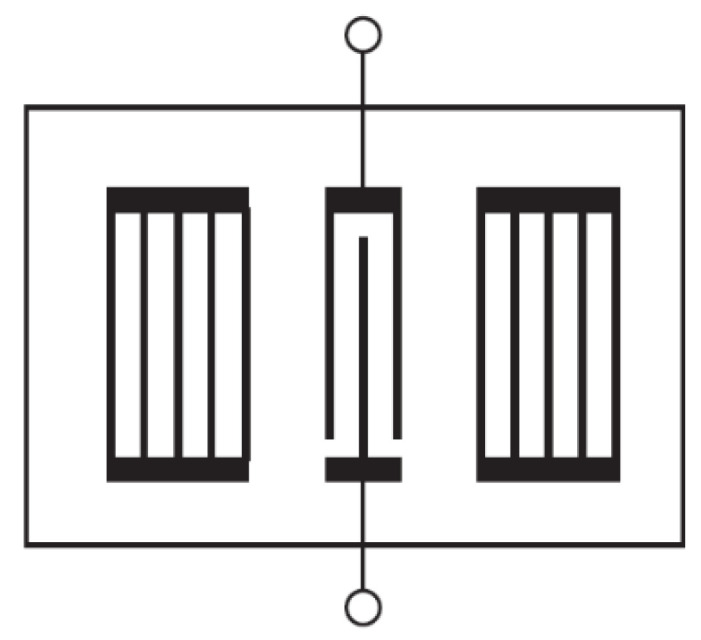
One-port SAW resonator with gratings, adapted with permission from [4].

**Figure 8 micromachines-14-00043-f008:**
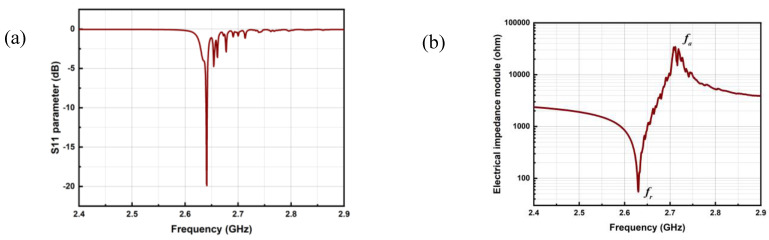
(**a**) *S*_11_ parameter and (**b**) electrical impedance of a 2.63 GHz ZnO based SMR (simulation results).

**Figure 9 micromachines-14-00043-f009:**
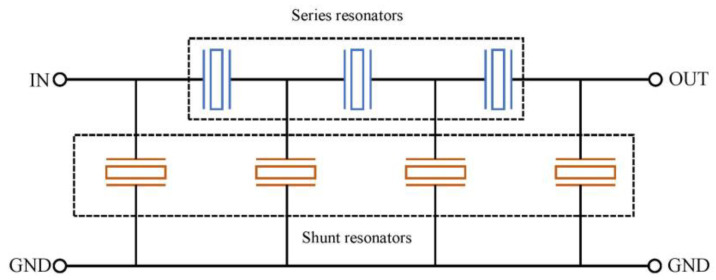
Ladder-type configuration filter for BAW and 1-port SAW devices.

**Figure 10 micromachines-14-00043-f010:**
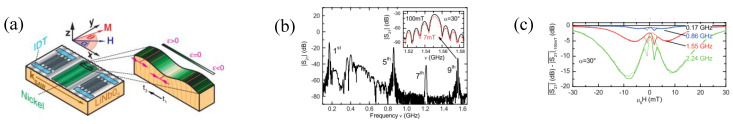
(**a**) Illustration of experimental setup; (**b**) *S*_21_ of the device as function of frequency when *H* = 0 and inset focused on the influence of external magnetic field *H* on the 9th harmonic; (**c**) *S*_21_ as function of *H* for several resonance modes, adapted with permission from [148].

## Data Availability

Not applicable.

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
