# Peer review of "Trends and Applications of Surface and Bulk Acoustic Wave Devices: A Review"

_micromachines, 2022, doi:10.3390/mi14010043_

Round 1

Reviewer 1 Report

It seems to me that Authors do not quite realise how expansive are SAW and BAW applications. The content is well written but far insufficient to be a review paper comprehensive enough. There is no information about: BAW filters, low loss SAW coupled filters, QCM sensors, SAW sensors for different physical quantities (e.g. gyroscopes), BAW and SAW actuators etc. 

It can be found many misleading information in the text. For instance, the sentence "Currently, two main structures of BAW resonators are commercially available: cav-145 ity type (FBAR) and Solidly Mounted type (Solidly Mounted Resonators, SMR)." is far from true, Figure 9 shows BAW ladder, actually - see e.g.:

https://www.rfwireless-world.com/Terminology/Difference-between-SAW-filter-types-IIDT-DMS-and-Ladder.html

Such a trend to simplyfy the technical draws have been observed, but this time Figure 9 may be misleading.

Unfortunately the paper needs major revision.

Author Response

Editor-in-Chief of Micromachines

Dear Editor and Reviewers,

Thanks for your time to review this manuscript. There are helpful comments and suggestions where we found the stimuli to improve our work. This document reports a discussion on the remarks made by the Reviewers and a brief summary of the changes introduced in the manuscript to address their concerns. Please note that we decided to revise the text of the manuscript in several places to improve the clarity of the exposition. The new version of the manuscript highlights in different colours for Reviewer 1 (Red).

The following are detailed explanations on how we have addressed these comments.

Reviewer 1 comments to the Author:  It seems to me that Authors do not quite realise how expansive are SAW and BAW applications. The content is well written but far insufficient to be a review paper comprehensive enough. There is no information about: BAW filters, low loss SAW coupled filters, QCM sensors, SAW sensors for different physical quantities (e.g. gyroscopes), BAW and SAW actuators etc. 

It can be found many misleading information in the text. For instance, the sentence "Currently, two main structures of BAW resonators are commercially available: cavity type (FBAR) and Solidly Mounted type (Solidly Mounted Resonators, SMR)." is far from true, Figure 9 shows BAW ladder, actually - see e.g.:

https://www.rfwireless-world.com/Terminology/Difference-between-SAW-filter-types-IIDT-DMS-and-Ladder.html

Such a trend to simplyfy the technical draws have been observed, but this time Figure 9 may be misleading.

Unfortunately, the paper needs major revision.

Comment#1

Reviewer wrote: It seems to me that Authors do not quite realise how expansive are SAW and BAW applications. The content is well written but far insufficient to be a review paper comprehensive enough.

Our answer:

Thank you for your comments which we understand and also share in another context.

Indeed, we have written and proposed this review with the aim of making this field of elastic wave devices accessible to novice readers such as students and young professionals. We would like to bring our readers to have a global and affordable understanding of what was done and what are the perspectives in this field, in an educational and concise way.

Moreover, in the literature, there are already many detailed and in-depth reviews on acoustic devices regarding the physics of acoustic device sensors[1], biosensors[2-6], chemical and gas sensors. [7, 8]. Furthermore, the basic theory, physics and design of filters are also well detailed in several books and articles [9-12]. Even with regard to quantum acoustics, many wonderful reviews can also be found in the literature [13-15].

Nevertheless, we’ve added some inputs here and there related to a broader application field, as described below. 

Comment#2

Reviewer wrote: There is no information about: BAW filters

Our answer:

Thanks for your comments that helped us to add information that we hope relevant about BAW filters from line 318 to line 342

Comment#3

Reviewer wrote: There is no information about: low loss SAW coupled filters.

Our answer:

Thanks for your comments, we are totally agreed with you. We therefore added some descriptions about low loss SAW coupled filters from line 291 to line 293 as following: << Similarly, a design coupling identical SAW resonators in double symmetric and anti-symmetric modes by inverting their electrical connections can result in a wider passband filter (additional references).>>

Comment#4

Reviewer wrote: There is no information about: QCM sensors and SAW sensors for different physical quantities (e.g. gyroscopes) and BAW and SAW actuators etc

Our answer:

We totally share your opinion and we are thankful for this relevant comment. We therefore added some descriptions about QCM sensors from the line 344 to line 352 as following: << Beside the filtering application, SAW and BAW devices are also widely used as sensors and different configurations have been developed for various physical quantities monitoring. Among them, the QCM has been used as a way of real-time monitoring thin film deposition thickness in microelectronics such as for evaporation equipment, based on mass-effect, with a resolution in the order of the ng.cm–2 (additional references). Many other applications have been studied, such as magnetic field (additional references), pressure and temperature (additional references), acceleration (additional references), stress, tire-road friction (additional references), gyroscope (additional references) in many sectors including automobile, consumer, etc (additional references). What’s more, SAW devices are also reported as motors and actuators (additional references). >>

Comment#5

Reviewer wrote: It can be found many misleading information in the text. For instance, the sentence "Currently, two main structures of BAW resonators are commercially available: cavity type (FBAR) and Solidly Mounted type (Solidly Mounted Resonators, SMR)." is far from true, 

Our answer:

Thanks for your comments. We made a change in this new version as following: <<The current FBAR devices can be divided into two main types by their structures: front side etch or air bag type and solidly mounted type (Solidly Mounted Resonators, SMR). >>. Other changes have been made to precise some sentences and avoid misleading information.

Comment#6

Reviewer wrote: Figure 9 shows BAW ladder, actually - see e.g.:

https://www.rfwireless-world.com/Terminology/Difference-between-SAW-filter-types-IIDT-DMS-and-Ladder.html. Such a trend to simplyfy the technical draws have been observed, but this time Figure 9 may be misleading.

Our answer:

Thanks for your comments. In fact, Figure 9 is inspired by some papers and published books.

Indeed, for SAW ladder-type filter, some similar illustrations can be found in [1] (Page 28, Figure 1.18 (a)), [2] (Page 154, Figure 5.38 (b)) and [3] (Page 1, Figure 2 (b)).

(a)

(b)

(c)

Fig 1(a) (SAW)Impedance element filter (IEF)[1]; (b) SAW Ladder-type filter[2]; (c) (SAW) Ladder type filter structure including bonding wire[3].

Of course, some similar illustrations can be found also for BAW ladder-type filter, such as in [4] (Page 98, Figure 7.1 (a)).

Fig 2 The topology of the FBAR filter: trapezoidal type (ladder type) [4]

In this context, we would like to simplify our illustration and we present both SAW and BAW ladder-type filter configuration like in Figure 9.

Reference

  1. Morgan, D., Surface acoustic wave filters: With applications to electronic communications and signal processing. 2010: Academic Press.
  2. Hashimoto, K.-y. and K.-Y. Hashimoto, Surface acoustic wave devices in telecommunications. Vol. 116. 2000: Springer.
  3. Ueda, M., et al. Low loss ladder type SAW filter in the range of 300 to 400 MHz. in 1994 Proceedings of IEEE Ultrasonics Symposium. 1994. IEEE.
  4. Zhang, Y. and D. Chen, Multilayer integrated film bulk acoustic resonators. 2012: Springer Science & Business Media.

Reviewer 2 Report

In this paper, the piezoelectric effect, basic principle, basic structure, application field and development prospect of RF acoustic devices (surface acoustic devices and bulk acoustic devices) are introduced by referring to a large number of literatures on acoustic devices. The performance and optimization methods of the devices are discussed. The novel application trends of high-frequency filters and sensors are discussed. Meaningful examples and applications are presented to support the reviews of this paper. In general, the paper has reference significance for systematically understanding the development process of elastic wave devices. While the current version is in good condition, the following minor modifications would make it even better.

 1.      The full names of bulk acoustic wave and surface acoustic wave appear many times in the paper, the bulk acoustic wave and surface acoustic wave should be written as BAW and SAW except the first time, carefully checked and corrected. 

2. Figure 2(a) is not detailed enough; it is better to describe the workflow.

3. In Figure. 3, the thickness of the piezoelectric layer h is not marked, where h is equal to λ/2?

4The quality factor (Q) value is mentioned many times in this paper. It is better to introduce it in detail, which can help beginners more. 

5. Section 2.1 introduces the basic structure of SAW and BAW resonators, and then discusses in detail the two main structures of BAW resonators (FBAR, SMR). Is there more structure of SAW resonators?

6. For an ideal BAW device, how do you know that the Q value can reach about 1200-1500? In reference 19, the BAW resonator based on a piezoelectric AlN film has a quality factor(Q)value of 2300. Figure 8 (b) should be marked clearly the corresponding resonance frequency ?? and anti-resonance frequency ??.

Author Response

Editor-in-Chief of Micromachines

Dear Editor and Reviewers,

Thanks for your time to review this manuscript. There are helpful comments and suggestions where we found the stimuli to improve our work. This document reports a discussion on the remarks made by the Reviewers and a brief summary of the changes introduced in the manuscript to address their concerns. Please note that we decided to revise the text of the manuscript in several places to improve the clarity of the exposition. The new version of the manuscript highlights in different colours for Reviewer 2 (Blue).

The following are detailed explanations on how we have addressed these comments.

Reviewer 2 comments to the Author:  In this paper, the piezoelectric effect, basic principle, basic structure, application field and development prospect of RF acoustic devices (surface acoustic devices and bulk acoustic devices) are introduced by referring to a large number of literatures on acoustic devices. The performance and optimization methods of the devices are discussed. The novel application trends of high-frequency filters and sensors are discussed. Meaningful examples and applications are presented to support the reviews of this paper. In general, the paper has reference significance for systematically understanding the development process of elastic wave devices. While the current version is in good condition, the following minor modifications would make it even better.

Comment#1

Reviewer wrote:

  1. The full names of bulk acoustic wave and surface acoustic wave appear many times in the paper, the bulk acoustic wave and surface acoustic wave should be written as BAW and SAW except the first time, carefully checked and corrected. 

Our answer:

Thanks for your comments and reminders. You are right, we have corrected them.

Comment#2

Reviewer wrote:

  1. Figure 2(a) is not detailed enough; it is better to describe the workflow.

Our answer:

Thanks for your comments and suggestions. The legend of Figure 2(a) is changed into ‘Schematic diagram of classic SAW resonator with delay-line’ More descriptions about Figure 2(a) are given in the text as below from line 88 to line 96:

…..In 1965, Richard Manning White et al. proposed to deposit directly interdigital transducers (IDTs) onto the surface of piezoelectric materials in order to generate, transmit and receive SAW efficiently(additional references). As represented in the Figure 2 (a), in a standard SAW resonator with delay-line, when the electrical signal arrives at the input IDT (left side) through feedline, here with matching dipole, acoustic waves are generated by inverse piezoelectric effect and acoustic resonance occurs, at specific frequencies of constructive waves, for which acoustic waves travel across the propagation path. Arriving at the output IDT (right side), the acoustic signal is therefore converted back into electrical signal through the output feedline by piezoelectric effect.

Comment#3

Reviewer wrote:

  1. In Figure. 3, the thickness of the piezoelectric layer h is not marked, where h is equal to λ/2?

Our answer:

Thanks to your comment, h is added into the Figure 3 for a better illustration.

Comment#4

Reviewer wrote:

4.The quality factor (Q) value is mentioned many times in this paper. It is better to introduce it in detail, which can help beginners more. 

Our answer:

Your comment is very helpful and indeed will make the definitions clearer. The definition and some brief descriptions about Q value are introduced in this new version from line 141 to line 142.

Comment#5

Reviewer wrote:

  1. Section 2.1 introduces the basic structure of SAW and BAW resonators, and then discusses in detail the two main structures of BAW resonators (FBAR, SMR). Is there more structure of SAW resonators?

Our answer:

Thanks for your comments. SAW devices can be divided into one port resonator (As shown in Figure 7) and two ports resonator with delay-line (As shown in Figure 2. (a)).  IDT can be divided into single and split-finger electrodes (As shown in Figure 6.). Other modified IDT structures like apodized type[1] (Page 18, Figure 1.11 (a)) or pulse compression type[1] (Page 6, Figure 1.5) are not introduced because we have added the citation of these book references [1] and [2] which detail all these definitions and will help the reader to deepen his curiosity if he wishes to go further. Besides, we’ve also introduced some SAW configurations with gratings, as well as SAW-based circuits such as the ladder-type and other coupled mode designs.

Comment#6

Reviewer wrote:

  1. For an ideal BAW device, how do you know that the Q value can reach about 1200-1500? In reference 19, the BAW resonator based on a piezoelectric AlN film has a quality factor(Q)value of 2300. Figure 8 (b) should be marked clearly the corresponding resonance frequency ??and anti-resonance frequency ??.

Our answer:

Thanks for your comments. Indeed, the Q value of 1200-1500 is gotten from our homemade performed simulation results with different designs. It is true that higher Q value can be found in the literature and more attentions should be paid. Thus, we modified the description into ‘For a BAW device with good performance, the Q value can reach several thousands (added citations).’ The ?? and anti-resonance frequency ?? are also added into Figure 8(b).

Reference

  1. Morgan, D., Surface acoustic wave filters: With applications to electronic communications and signal processing. 2010: Academic Press.
  2. Hashimoto, K.-y. and K.-Y. Hashimoto, Surface acoustic wave devices in telecommunications. Vol. 116. 2000: Springer.

Reviewer 3 Report

The paper is of interest to people who are completely new to SAW/BAW and who are interested in RF filter and/or liquid and gas phase sensor applications. It also compiles some interesting information, concerning new R&D topics (e.g., SAW for quantum acoustics) and the state of the world market for SAW/BAW filters. So, it is a potentially useful paper, to get started.

The paper is well written, with care. However, there are still some small mistakes, to be corrected (typos).

Author Response

Editor-in-Chief of Micromachines

Dear Editor and Reviewers,

Thanks for your time to review this manuscript. There are helpful comments and suggestions where we found the stimuli to improve our work. This document reports a discussion on the remarks made by the Reviewers and a brief summary of the changes introduced in the manuscript to address their concerns. Please note that we decided to revise the text of the manuscript in several places to improve the clarity of the exposition. The new version of the manuscript highlights in different colours for Reviewer 3 (Green).

The following are detailed explanations on how we have addressed these comments.

Reviewer 3 comments to the Author:  The paper is of interest to people who are completely new to SAW/BAW and who are interested in RF filter and/or liquid and gas phase sensor applications. It also compiles some interesting information, concerning new R&D topics (e.g., SAW for quantum acoustics) and the state of the world market for SAW/BAW filters. So, it is a potentially useful paper, to get started.

The paper is well written, with care. However, there are still some small mistakes, to be corrected (typos).

Comment#1

Reviewer wrote:

  1. The paper is of interest to people who are completely new to SAW/BAW and who are interested in RF filter and/or liquid and gas phase sensor applications. It also compiles some interesting information, concerning new R&D topics (e.g., SAW for quantum acoustics) and the state of the world market for SAW/BAW filters. So, it is a potentially useful paper, to get started.

Our answer:

Thank you for your comments which we understand and also share in another context.

Indeed, we have written and proposed this review with the aim of making this field of elastic wave devices accessible to novice readers such as students and young professionals. We would like to bring our readers to have a global and affordable understanding of what was done and what are the perspectives in this field, in an educational and concise way.

Moreover, in the literature, there are already many detailed and in-depth reviews on acoustic devices regarding the physics of acoustic device sensors[1], biosensors[2-6], chemical and gas sensors. [7, 8]. Furthermore, the basic theory, physics and design of filters are also well detailed in several books and articles [9-12]. Even with regard to quantum acoustics, many wonderful reviews can also be found in the literature [13-15].

Finally, based on the other reviewers’ comment, we added more information to make this article more relevant.

Comment#2

Reviewer wrote:

  1. The paper is well written, with care. However, there are still some small mistakes, to be corrected (typos).

Our answer:  Thanks for your comments. Some mistakes have been detected and corrected.

Round 2

Reviewer 1 Report

The range of changes is insufficient just at first glance.

I gave already detailed remarks concerning necessary corrections. Seems to me most of them were ignored.

Author Response

Editor-in-Chief of Micromachines

Dear Editor and Reviewers,

Thanks for your time to review this manuscript. We took more time for an additional deep revision in order to better fit the helpful comments and suggestions from the reviewer and improve the manuscript accordingly. This document reports on the changes done and especially additional material to address these concerns and, we hope, clarify the content and objectives of this review. The new version of the manuscript highlights these changes (yellow colour). The fact is that, the expansive SAW and BAW applications lead to a profusion of associated literature, and this article particularly aims to help the young professionals and students to get a comprehensive overview of such acoustic technologies and of major applications, with a focus on emerging ones, without being exhaustive and with references to the literature for those who will want to deepen a particular topic. Beyond the additional material included in the new manuscript, we have introduced this aim in the abstract and the introduction in order to clarify. More generally, both the abstract and the introduction, as well as the concluding part, have been modified to reflect the content changes.

Please find below the explanations and major changes done to address the reviewer 1 comments.

Reviewer 1 comments to the Author (Round 2):  

The range of changes is insufficient just at first glance.

I gave already detailed remarks concerning necessary corrections. Seems to me most of them were ignored.

Our general answer:

We would like to really thank you for your comments that we considered with great attention. This new revision includes deep changes, among them a full sub-part dedicated to the applications of acoustic devices as physical sensors and actuators, as well as several additional paragraphs related to BAW filters and QCM sensors. Please find our reply to your previous comments in below. Furthermore, the fact is that, the expansive SAW and BAW applications lead to a profusion of associated literature, and this article particularly aims to help the young professionals and students to get a comprehensive overview of such acoustic technologies and of major applications, with a focus on emerging ones, without being exhaustive and with references to the literature for those who will want to deepen a particular topic. In order to clarify this, beyond the additional material specifically associated to your suggestions included in the new manuscript, we have also introduced this aim in the abstract and the introduction. More generally, both the abstract and the introduction, as well as the concluding part, have been modified to reflect the content changes.

In below, we give more details about how we specifically answered your comments (Round 1), to evidence and explain the major changes done for each part of them and especially about the material added to address these concerns.

In the new version of the manuscript the changes are highlighted in yellow. We hope this will clarify the content and objectives of the article.

Reviewer 1 comments to the Author (Round 1):

It seems to me that Authors do not quite realise how expansive are SAW and BAW applications. The content is well written but far insufficient to be a review paper comprehensive enough. There is no information about: BAW filters, low loss SAW coupled filters, QCM sensors, SAW sensors for different physical quantities (e.g. gyroscopes), BAW and SAW actuators etc.

It can be found many misleading information in the text. For instance, the sentence "Currently, two main structures of BAW resonators are commercially available: cavity type (FBAR) and Solidly Mounted type (Solidly Mounted Resonators, SMR)." is far from true, Figure 9 shows BAW ladder, actually - see e.g.:

https://www.rfwireless-world.com/Terminology/Difference-between-SAW-filter-types-IIDT-DMS-and-Ladder.html

Such a trend to simplyfy the technical draws have been observed, but this time Figure 9 may be misleading.

Unfortunately, the paper needs major revision.

Comment part#1

Reviewer wrote: There is no information about: BAW filters

Our answer: Thanks for your comments that helped us to add information that we hope relevant about BAW filters from line 331 to line 342 as following: << Like SAW, BAW device, and currently increasingly of FBAR-type, is also a funda-mental component for RF filters, requiring a wide bandwidth with low insertion loss and a stopband enabling the suppression of unwanted signals. Similarly, as for the SAW components, the ladder type filters are also commonly used due to advantages such as easy design, steep filtering effect, cost, etc. Some communication systems mix ladder filters and lattice ones, in which shunt elements are diagonally-crossed, to achieve a good selectivity of frequency band with steeper filtering response [26, 45]. Since the resonance frequency of SAW mainly depends on the spatial periodicity of IDTs, which is limited by lithography and patterning technology, it is quite difficult for SAW to operate above 2 GHz [46], so that FBAR usually dominates the market for fil-ters above 2.5 GHz. This is further reinforced with the 5G New Radio (NR) systems, since a main feature is the use of high-frequency millimeter wave (mmWave) and sub-6 GHz bands [47].>>

Comment part#2

Reviewer wrote: There is no information about: low loss SAW coupled filters.

Our answer: Thanks for your comments, we totally agree with you. We therefore completed the first paragraph of RF filters in order to explicitly include some descriptions about low loss SAW coupled filters, from line 293 to line 305 << In this perspective, as for unidirectional IDTs, an appropriate design can generate spe-cific filter templates [4]. Beyond that, much attention has also been paid to filter con-figurations that smartly combine several one port-SAW resonators, as impedance ele-ments, with different topologies like Inter-digitated Inter-Digital Transducer SAW (IIDT), Double Mode SAW (DMS) or ladder type [41]. The ladder-type filter is a very common configuration of such low-loss coupled SAW, represented in the Figure 9, which consists of cascaded multiple stages, each one based on two SAW resonators connected in series and in parallel. Coupling them by matching the resonance fre-quency of a serial resonator with the anti-resonance frequency of the parallel one re-sults in a bandpass filter centered on this frequency. Such filters exhibit a relatively flat passband with low loss and good rejection of out-of-band noise [30]. Similarly, a design coupling identical SAW resonators in double symmetric and anti-symmetric modes by inverting their electrical connections can result in a wider passband filter [30, 42].>>

Comment part#3

Reviewer wrote: There is no information about: QCM sensors

Our answer: Thanks for your comments. We added some description about QCM sensors from line 477 to line 485 as following: << Similarly, BAW devices are also widely used as chemical and biological sensors, firstly based on the classical QCM, and taking into account surrounding physical pa-rameters such as temperature and pressure as previously described. Its operating fre-quency can go up to tens of MHz and thickness shear mode (TSM) is mostly studied [86]. With similar advantages than SAW devices except a lower frequency, some commercial products based on QCM allowed a large use for a number of applications involving mass changes measurements at the nanoscale resolution. Moreover, the ad-dition of dissipation monitoring, known as “QCM-D”, allows an improved characteri-zation of both mass and viscoelastic properties changes of the medium driven at the near surface [118].>>

Comment part#4

Reviewer wrote: There is no information about: SAW sensors for different physical quantities (e.g. gyroscopes) and BAW and SAW actuators etc

Our answer: We totally share your opinion and we are thankful for this relevant comment. In this new revision, we added a full new section dedicated to physical sensors and actuators from the line 363 to line 415.

Some acoustic wave sensors market information is also added from the line 537 to line 544 as following: << As we presented in above, acoustic wave devices (SAW and BAW)-based sensors have wide applications as physical and bio-chemical sensors, the market of these sen-sors also hold an important share. From the report “Acoustic Wave Sensor Market - Forecasts from 2021 to 2026” by Research and Markets, global acoustic wave sensor market will grow from US$ 836.17 million in 2019 to US$ 2400.54 million in 2026, with a CAGR of 16.3%. The major players are Hawk Measurement Systems (Australia), NanoTemper technologies GmbH (Germany), Pro-micron GmbH (Germany), Siemens (Germany) and Transense (UK) [135].>>

As mentioned before, some other additional changes have been made, especially (but not exclusively) in the abstract, the introduction and in the concluding part, in order to reflect the changes in the text and aiming to complete the descriptions and also clarify the content and objectives.

Finally, we would like to thank you again for your highly helpful comments and we hope that we succeeded in improving the manuscript accordingly in this new revision.

On behalf of the authors,
Yang YANG

Round 3

Reviewer 1 Report

The text is little bit better now. It describes a chosen trends and applications of BAW and SAW devices. Actually, it is small part of large area of the devices technology. Still to small to be comprehensive review. 

Perhaps the term "chosen" is the way to fit the title to the content of the paper.